# Genome-Wide Analysis and Profile of UDP-Glycosyltransferases Family in Alfalfa (*Medicago sativa* L.) under Drought Stress

**DOI:** 10.3390/ijms23137243

**Published:** 2022-06-29

**Authors:** Bao Ao, Yangyang Han, Shengsheng Wang, Fan Wu, Jiyu Zhang

**Affiliations:** State Key Laboratory of Grassland Agro-Ecosystems, Key Laboratory of Grassland Livestock Industry Innovation, Ministry of Agriculture and Rural Affairs, College of Pastoral Agriculture Science and Technology, Lanzhou University, Lanzhou 730020, China; aob20@lzu.edu.cn (B.A.); hanyy20@lzu.edu.cn (Y.H.); shshwang21@lzu.edu.cn (S.W.); wuf15@lzu.edu.cn (F.W.)

**Keywords:** *Medicago sativa* L., UDP-glycosyltransferases, drought stress response

## Abstract

Drought stress is one of the major constraints that decreases global crop productivity. Alfalfa, planted mainly in arid and semi-arid areas, is of crucial importance in sustaining the agricultural system. The family 1 UDP-glycosyltransferases (UGT) is indispensable because it takes part in the regulation of plant growth and stress resistance. However, a comprehensive insight into the participation of the UGT family in adaptation of alfalfa to drought environments is lacking. In the present study, a genome-wide analysis and profiling of the UGT in alfalfa were carried out. A total of 409 UGT genes in alfalfa (*MsUGT*) were identified and they are clustered into 13 groups. The expression pattern of *MsUGT* genes were analyzed by RNA-seq data in six tissues and under different stresses. The quantitative real-time PCR verification genes suggested the distinct role of the *MsUGT* genes under different drought stresses and abscisic acid (ABA) treatment. Furthermore, the function of *MsUGT003* and *MsUGT024*, which were upregulated under drought stress and ABA treatment, were characterized by heterologous expression in yeast. Taken together, this study comprehensively analyzed the UGT gene family in alfalfa for the first time and provided useful information for improving drought tolerance and in molecular breeding of alfalfa.

## 1. Introduction

Drought is one of the main environmental stresses affecting plant growth and development process causing significant reduction in crop yield and quality. The threat of drought to agriculture systems is aggravated because of the reduction of fresh water resources and increasing food demand [1]. A decrease in water availability would have a deleterious effect on plant growth, because about 80–95% of plant fresh weight is comprised of water [2]. While plants have evolved the ability to alter biological processes to avoid the harm of stress, under drought condition, the plant stress-protectant metabolites increased and the antioxidant system activated to maintain redox homeostasis [3]. Activated drought stress pathways also includes phytohormones, such as abscisic acid (ABA), which serves as a first stress signal to promote stomatal closure to induce osmotic adjustment and alter gene expression to accommodate to conditions of water defecit [4].

Glycosylation is a pronounced and universal modification found in all living systems, which mainly affects the transport, stability, storage, reactivity, and bioactivity of the sugar acceptors [5]. Glycosyltransferases (GTs), an unusually large enzyme family, were classified into 114 superfamilies (CAZy, http://www.cazy.org, accessed on 20 April 2021) depending on similarities of amino acids, substrate specificity, catalytic functions, and the existence of conserved sequence motifs. Among them, the largest glycosyltransferase family in plant species, GT family-1, also named uridine diphosphate glycosyltransferase (UGT), catalyzes the covalent addition of sugars from nucleotide UDP-sugar donors to operative groups such as carboxyl, hydroxyl, and amine on a wide variety of lipophilic molecules [6]. The UGTs are usually identified through the consensus sequence called the plant secondary product glycosyltransferase box (PSPG box) at the C-terminal end, which is involved in the binding of acceptor substrates with the enzyme [7]. The PSPG box is a consensus sequence constituted of 44 amino acids residues and occurs in all plants [8]. The UGTs encompass a vital role in metabolic homeostasis, such as magnification of water solubility, deactivation, and elimination of toxic products [9]. The UGTs widely participate in the biosynthesis of secondary metabolites, thus regulating the plant growth, development and in response to abiotic and biotic stress [10,11].

To date, the role of UGTs has been elucidated in numerous plant species, animals, fungi, and bacteria [12]. One hundred twenty *UGT* gene members were first identified in the model plant *Arabidopsis thaliana* and classified into 14 groups [13]. Owing to the advancements in the high-throughput sequencing of plants in recent years, a large number of UGT genes were identified and characterized in *Zea mays* [14], *Oryza sativa* [15], *Triticum aestivum* [16], *Brassica rapa*, *Brassica napus* [17], *Trifolium paratense* [18], *Melilotus albus* [19] and *Medicago truncatula* [18]. The prevalence of the UGTs in plants indicates their vital roles in biosynthesis and stress resistance. The UGTs could potentially impact the life cycle of plants by regulating their growth and development. Previous studies have shown that several *UGT* genes play notable roles in modifying plants’ secondary metabolites that assist the plants to adapt to changing environments; for example, *UGT76B1* and *UGT03300* have roles in plant defense to pathogens [20,21], and *UGT85A5* in tolerating salt stress [22]. Under conditions of water defecit, the ABA-specific glycosyltransferase *UGT71B6* in *A. thaliana* can glycosylate ABA to conjugate with glycosyl ester (GE), producing physiologically inactivated ABA-GE, probably a storage form of ABA in vacuoles, which reduces water loss from leaves [23], while *UGT71C5* was demonstrated to glycosylate ABA in vitro and in vivo, and mutation or down-expression of this gene improved drought resistance in *A. thaliana* [24]. *SIUGT75C1* from tomato, which catalyzed the glycosylation of ABA, was demonstrated to be involved in fruit ripening and drought responses [25].

Alfalfa (*Medicago sativa* L.) is a prominent forage crop with high nutritional quality and yield. It plays a critical role in supplying livestock with highly nutritious feed and strengthening soil fertility through biological N fixation [26]. Alfalfa is grown in more than 80 countries exceeding 35 million hectares all around the world [27]. However, in these areas, water deficit is one of the main constraints that reduces alfalfa yield and quality. Therefore, it is vital to improve adaptation of alfalfa to reduced water allocations and to rainfed conditions [12]. A noteworthy characteristic of alfalfa is its plasticity in terms of its adaptation to complicated and changeable environmental and climatic conditions [28]. The release of chromosome-level genome assembly with 32 chromosomes of alfalfa cultivar Zhongmu-4 has provided abundant data resources for selecting drought-related genes and improving alfalfa drought resistance [29]. However, a comprehensive insight into the participation of the UGT family in adaptation of alfalfa to drought environments is lacking. We hypothesized that glycosyltransferases play a notable role in coping with drought stress and interacting with ABA in alfalfa. In the present study, we identified 409 UGT genes from alfalfa depending on the accomplishable alfalfa genome. The genetic relationships of these UGT genes were confirmed by sequencing alignment and analyzing the phylogenetic tree. The chromosome locations, proteins’ structure, motifs’ composition cis-regulatory and synteny of the MsUGTs were conducted with the assistance of the web services. The expression patterns of *MsUGT* genes in six different tissues and responses to different abiotic stresses were investigated. We further characterized two genes in vitro to confirm their roles in drought stress and ABA treatments. Our study is expected to contribute to conducting functional characterization analysis of alfalfa *UGT* genes to better understand the molecular bases of responses to drought tolerance.

## 2. Results

### 2.1. Identification of UGT Genes in Alfalfa

A recent complement of high-quality assembly of the chromosome-level genome sequence of Zhongmu-4, which combines both illumines and Pacbio sequencing data, enabled an in-depth analysis of *UGT* genes in alfalfa [30]. According to blast search and gene annotation of Pfam and NCBI databases, putative *UGT* genes were obtained. After removing redundant genes or those lacking the PSPG box, a total of 409 genes were identified as *UGT* genes in alfalfa. For convenience, these genes were named *MsUGT001* to *MsUGT409* according to the physical distribution of genes on the chromosomes. Most of the genes ranged from 300 to 500 amino acids, except for a few genes that were above 800 and below 200 amino acids. The theoretical isoelectric point (pI) ranged from 4.77 (*MsUGT183*) to 9.91 (*MsUGT133*) with an average of 5.80. The molecular weight (Mw) varied between 14,478.83 (*MsUGT133*) and 248,800.02 (*MsUGT401*) with an average of 57,118.89 Da. The prediction of cellular localization of 409 *MsUGT* genes showed that 316 and 173 genes were localized in the cytoplasm and plasma membrane, respectively. Besides, 81 genes were predicted to be localized both in the plasma membrane and the cytoplasm, and 20, 21,1 and 4 genes were distributed in the chloroplast, nuclear, extracellular, lysosomal, and mitochondrial, respectively. (Appendix A).

### 2.2. Phylogenetic Analysis of MsUGTs

Based on the 17 *A. thaliana* and 14 *M. truncarula* UGTs sequences the phylogenetic analysis of *UGT* genes in alfalfa was conducted (Figure 1). The phylogenetic trees of UGT members were clustered into 17 groups, which were named A–N (based on identified group in *A. thaliana*)*;* and O, P, R that were newly found in *M. truncatula* and alfalfa. The MsUGTs were clustered into 13 groups, lacking *A. thaliana* 4 conserved phylogenetic groups (A, G, J, N), and groups O, P, and R were found in alfalfa. The number of UGT members in each group was inconstant: the largest group was I, which contained 134 gene members, and the smallest two groups were C and R, both of which contained only 2 members. Groups L, K, H, M, E, F, B, and D had 43, 10, 13, 9, 62, 6, 4, and 100 members, respectively. In addition, the two newly identified groups O and P contained 10, and 14 members, respectively.

### 2.3. Genomic Localization and Synteny Analysis of MsUGT Genes

The genetic localization on the chromosome was mapped based on the newly released Zhongmu-4 genome annotation information. Three hundred and eighty-one genes out of 409 *MsUGT* genes were randomly located in 32 chromosomes, another 28 *MsUGT* genes were not distributed to any chromosomes as anchored on the scaffolds. There were high densities of *MsUGT* genes clustered at a particular site on chromosomes 6 and 7 (Figure 2). The number of *MsUGT* genes varied from a minimum of 15 in chromosome 2 to a maximum of 118 genes on chromosome 6. Within them, chr2_1, chr2_2, chr2_3 and chr2_4 had 3, 7, 3 and 2 genes, respectively, while chr6_1, chr6_2, chr6_3 and chr6_4 had 33, 25, 30 and 30 genes, respectively. In addition, there were 22, 25, 16, 58, 81 and 61 *UGT* genes located in chromosome1, 3, 4, 5, 7 and 8 respectively.

Gene duplication is regarded as one of the fundamental driving pressures that contributes to the genomic evolution as well as the genetic systems [31]. Therefore, we executed collinearity analysis to identify the gene duplication events of *UGT* genes in alfalfa. Seven-hundred and forty-six pairs of genes were identified to have duplicated segments (Figure 3), among them 122 pairs had tandem duplication events (Appendix A). Interestingly, those *UGT* genes mainly from group L did not have duplicated events.

### 2.4. Conserved Motifs and Gene Structure of MsUGT Genes

It is well known that conserved protein motifs are essential to protein function and exon-intron structure is crucial to gene regulation [32]. Therefore, the distribution of conserved motifs and exon-intron were investigated systematically in alfalfa UGT proteins. A total of 20 motifs were verified in the 409 UGT proteins. Motif 1, which was the PSPG domain, and motifs 2 and 3 were observed in all of the UGT proteins. The distribution of the 20 motifs in each UGT members is shown in Appendix A. As expected, members in the same subfamily had parallel motifs; for example, group I had motif 4, 8, and 18, while other groups did not have them. Motif 20 existed only in groups E, F, and D. Besides, Motif 3 was in the beginning and motif 11 and 7 were in the tail of most of the sequences. Further, the exon-intron structures of *MsUGT* genes were displayed to elucidate the diversity of structure of these genes, (Appendix A). Out of the 409 identified genes, 269 genes (65.8%) contained introns, and 165 genes (40.3%) contained UTR. The number of introns varied from 1 to 27 in those genes, 168 genes contained only 1 intron and 8 genes contained more than 10 introns.

### 2.5. Cis-Regulatory Elements Analysis in the MsUGT Gene Promoters

The transcriptional regulation of the *MsUGT* genes was analyzed by predicting potential cis-regulatory in upstream promoter regions (2000 bp) by online service PlantCARE. A total of 22 cis-elements were recorded, involving abiotic stress, hormones, light response and developmental regulation (Appendix A). A total of 800 ABRE cis-elements involved in ABA, and 230 TC-rich repeat cis-elements related to defense and stress response were identified (Appendix A). Besides, 258 *MsUGT* genes contained more than 10 cis-regulatory elements in their promoter regions, indicating the *MsUGT* genes are likely to respond to various stress responses and in the regulation of synthesis of secondary metabolites.

### 2.6. Expression Pattern of MsUGT Genes in Six Different Tissues

To access the expression pattern of *MsUGT* genes in different tissues, the microarray datasets of flower, nodule, leaf, root, elongating stem internodes and post-elongation stem internodes of alfalfa were downloaded, based on the previous study. The RNA-seq datasets showed that 134 *MsUGT* genes were expressed in all six tissues and 407 genes were expressed in at least one tissue (Figure 4). Among these genes, 99, 60, 22, 28, 24 and 10 *MsUGT* genes were highly expressed (FPKM > 10) in flower, leaf, root, nodule, elongating stem internodes and post-elongation stem internodes, respectively, indicating diversified glycosyltransferases processes function in different tissues. Besides, there were no links between the expression pattern and the phylogenetic groups, suggesting the expression pattern of each *MsUGT* gene was unique.

### 2.7. Expression Pattern Analysis of MsUGT Genes under Abiotic Stresses and ABA Treatments

The expression levels of the *MsUGT* genes under abiotic stresses and ABA treatments were analyzed using the RNA-seq data downloaded from NCBI (Appendix A). The results showed that 384 (93.89%) *MsUGT* genes were expressed in at least one stress condition. In addition, 109 (26.65%), 134 (32.76%), 166 (40.59%), 180 (44.01%) and 167 (40.83%) *MsUGT* genes were highly expressed (FPKM ≥ 1) in low temperature, ABA, drought stress, high temperature and salt stress, respectively.

### 2.8. Relative Water Content of Leaves under Drought Stress and ABA Treatments

We measured the relative water content of leaves (RWC) under ABA treatments, PEG-induced drought stresses, respectively. The RWC of leaves showed no significant differences among 0 h, 24 h and 48 h under ABA treatments. While, the RWC of leaves significantly decreased 48 h after 15% and 20% PEG conditions, respectively. The RWC of leaves showed no significant difference 24 h after 15% PEG condition. Therefore, we defined 15% PEG and 20% PEG after 48 h as mild drought (MD) and severe drought (SD), respectively (Figure 5).

### 2.9. qRT-PCR Analysis of MsUGT Genes under Drought Stress and ABA Treatments

Then *MsUGT* genes in different subfamilies (*MsUGT003*, *MsUGT024*, *MsUGT028*, *MsUGT045*, *MsUGT091*, *MsUGT100*, *MsUGT113*, *MsUGT279*, *MsUGT280*, *MsUGT305*, *MsUGT359*, *MsUGT386*), which potentially responded to drought stress and ABA, were chosen for qRT-PCR analysis to confirm their expression profile under different levels of drought stresses and exogenous ABA treatment. The results showed that the expression levels of the *MsUGT* genes varied greatly in shoots and roots under different drought stresses and ABA treatments (Figure 6). The expression of 4 genes (*MsUGT024*, *MsUGT045*, *MsUGT100* and *MsUGT305*) were upregulated both in shoot and root upon exposure to mild and severe drought stresses, indicating positive regulation. In addition, *MsUGT028* was upregulated in the shoot and downregulated in the root, while, *MsUGT113* and *MsUGT279* were upregulated in the root and downregulated in the shoot when exposed to mild and severe drought stresses. For ABA treatment, the expression pattern of these genes varied greatly: *MsUGT003* and *MsUGT305* were upregulated in shoot and root under ABA treatment, and *MsUGT045*, *MsUGT091*, *MsUGT279*, *MsUGT359* and *MsUGT386* were downregulated. Besides, 4 genes (*MsUGT024*, *MsUGT028*, *MsUGT100*, *MsUGT113*) were upregulated in root and downregulated in the shoot, while only one gene (*MsUGT280*) was upregulated in shoot and downregulated in root when exposed to ABA treatment.

### 2.10. MsUGT003 and MsUGT024 in Response to Drought Tolerance and ABA Treatment in Yeast

*MsUGT003* and *MsUGT024* were significantly upregulated under different levels of drought stresses and ABA treatments. Herein, we assumed that the two representative genes contributed to coping with drought stress and ABA signaling in alfalfa. The function of *MsUGT003* and *MsUGT024* in response to drought and ABA treatments were investigated in transformed yeasts using the pYES2−*MsUGT003* and pYES2−*MsUGT024* constructs (Figure 7). The results indicated that there was no difference among empty vector lines and transformed lines except 10^6^-fold dilution under control condition. The growth of *MsUGT003* transformed yeast and the empty vector line were significantly influenced by 250 μM ABA treatment, while the *MsUGT024* transformed yeast continued growing under 10^4^ and 10^5^-fold dilution. On the other hand, under 30% PEG condition, *MsUGT024* transformed yeasts showed more sensitivity to drought than empty vector yeasts, but *MsUGT003* transformed yeasts depicted resistance to drought stress, especially under 10^6^-fold dilution.

## 3. Materials and Methods

### 3.1. Identification of UGT Genes in Alfalfa

The members of the *UGT* genes were identified in accordance with the alfalfa genome assemble of Chinese cultivar Zhongmu4 by local BLAST (2.6.0, Madden, T. L., Bethesda, USA) search with *A. thaliana*, *O. sativa* and *M. truncatula UGT* sequences as queries (e-value cut-off >1 × 10^−5^). The *UGT* sequences of *A. thaliana*, *O. sativa* and *M. truncatula* were accessed from the phytozome website (https://phytozome.jgi.doe.gov/pz/portal.html, accessed on 4 May 2021). The potential protein sequences of UGT members of alfalfa were further identified by Pfam (http://pfam.xfam.org/search#tabview=tab1, accessed on 14 May 2021) and NCBI (https://www.ncbi.nlm.nih.gov/Structure/cdd/wrpsb.cgi, accessed on 14 May 2021) databases to confirm the presence of the UDP-glycosyltransferases protein sequences. The isoelectric point (pIs), molecular weight, and amino acid amount were predicted with the online service ExPASy Protparm tool (https://web.expasy.org/protparam/, accessed on 20 May 2021) [33]. The subcellular locations of UGT proteins were forecasted by CELLO v2.5 (http://cello.life.nctu.edu.tw/, accessed on 20 May 2021) [34].

### 3.2. Multiple Sequences Alignment and Phylogenetic Tree Construction

Diverse alignments of the identified amino acid sequences of *M. sativa* as well as 17 *A. thaliana* and 14 *M. truncarula* UGTs were conducted using the Clustal X program with the default settings. The phylogenetic tree was carried out by the MEGA 7 using Neighbor-Jointing methods with the following settings: bootstrap values were set as 1000 replicates along with p-distance methods were set as pairwise deletion options for dealing with gaps among the amino acid sequences [35]. The polygenetic tree was displayed by the online program iTOL (https://itol.embl.de/, accessed on 25 May 2021) [36].

### 3.3. Chromosomal Locations of the MsUGT Genes and Gene Duplication Analysis

The allocation of the *MsUGT* genes on chromosomes was displayed by the online program MG2C (http://mg2c.iask.in/mg2c_v2.1/, accessed on 25 May 2021) laid on the genome annotation files of *M. sativa* [37]. Duplication events of *MsUGT* genes were characterized by the MCScanX and drawn by TBtools software [38].

### 3.4. Gene Structure and Motifs’ Composition of the MsUGTs

The *MsUGT* gene structures were displayed by the online tool Gene Structure Display Server (http://gsds.gao-lab.org/, accessed on 3 June 2021) using the coding sequences and the genome sequences [39]. The consensus motifs of MsUGTs were identified through the online MEME server using the deduced amino acid sequences (http://meme-suit.org/, accessed on 3 June 2021) [40] with the subsequent settings: the site distribution was set as any number of repetitions and the number of motifs was 20; the minimum motif sites and width were 5 and 6, respectively; and maximum motif sites and width were both set as 100.

### 3.5. Cis-Regulatory Element Analysis

The sequence of 2000 bp from the promoters of the *MsUGT* genes were extracted using Tbtools. Cis-regulatory elements in the 2000 bp regions of *MsUGT* genes were characterized using the online service (http://bioinformatics.psb.ugent.be/webtools/plantcare/html/, accessed on 3 June 2021) [41].

### 3.6. Plant Materials, Relative Water Content, Drought and ABA Treatment

Seeds of the M. sativa L. cv. Gongnong No.1 were surface sterilized by sodium hypochlorite for 3 min and flushed with sterile water 3 times, then these seeds were put into Petri dishes with moistened filter papers for germination. After 3 days, uniform seedlings were transferred into hydroponic units containing 1/2 MS (Murashige & Skoog) Medium in a greenhouse. The conditions in the greenhouse are described below: 16 h light, 25 °C/8 h dark, 20 °C cycle, 75% relative humidity. Drought stress was applied to one-month-old seedlings using polyethylene glycol 6000 (PEG-6000): mild drought (MD) and severe drought (SD) conditions were created by using 15% and 20% PEG-6000, respectively; and the control condition (CK) was created using sterile water. ABA treatment was implied with 100 μM ABA solution sprayed onto the whole plant. Three young leaves were weighted immediately to obtain fresh weight after 0 h, 24 h and 48 h under MD, SD and ABA treatment, respectively. Then turgid weights (TW) were measured 5 h after putting the leaves into distilled water. Then leaves were dried in an oven at 75 ℃ 24 h to obtain dry weights (DW). Relative water content was calculated as the following formula. Young leaves and roots were sampled before treatments as control and after 48 h after treatments and flash frozen in liquid nitrogen. Thereafter these were stored at −80 °C for RNA preparation. All experiments were conducted for three biological replicates.
RLW = FW − DW/TW − DW × 100

### 3.7. RNA seq and qRT-PCR Analysis

The RNA-seq data of MsUGT genes were accessed from the NCBI (http://www.ncbi.nlm.nih.gov/sra, accessed on 10 July 2021) to investigate the expression pattern of different tissues (flower, nodule, root, leaf, elongation stem internodes and post-elongation stem internodes, SRP055547) and different abiotic stresses (drought stress (SRR16068779-83, SRR16068789-90), ABA (SRR7166039-40, SRR71660320-21), salt stress (SRR14999928-33), low temperature (SRR9888362-67) and high temperature (SRR10166266-69, SRR10166274-75)). The nucleotide sequences of all MsUGT genes were blasted against transcriptome datasets. Then the expression values and heatmap of MsUGT genes in different tissues and under different stresses were conducted by Tbtools. Twelve genes from different subfamilies with high FPKM values in the drought stress and ABA were selected for quantitative real-time PCR (qRT-PCR) experiments. RNA was extracted from shoot and root tissues after PEG-6000 or ABA treatment in accordance with the instruction provided with the RNAiso reagent (Takara, Dalian, China). The first-stand cDNA was reverse-transcribed from the extracted RNA which was separated from the whole genomic DNA using the TaKaRa reaction Kit. The qPCR was conducted using a SYBR Green qPCR kit (Sangon, Shanghai, China) in accordance with the manufacturer’s instruction on a CFX96 Real-Time PCR Detection System (Bio-Ras, Los Angeles, CA, USA). qPCR was performed in a 10 μL system containing 5 μL of 2 × SG Fast qPCR Master Mix, 0.2 μL of forward and reverse primers (10 μM each), 1 μL of DNA Buffer, 1 μL of cDNA, and 2.6 μL of double-distilled water. The relative expression level of each MsUGT gene was determined according to the 2^−ΔΔCt^ method [42]. The primers were designed using SnapGene software with melting temperatures ranging between 58 and 65 °C and synthesized by Sangon Biological Engineering Technology (Shanghai, China). Each biological replication was supported by three technical replicates.

### 3.8. Heterologous Expression Validation in Yeast

The pYES2−*MsUGT003* and pYES2−*MsUGT024* were constructed according to the previous study [19]. Briefly, the full-length coding sequences of *MsUGT003* and *MsUGT024* were reverse-transcribed from the RNA using a ClonExpress^®^ MultiS One Step Cloning Kit (Vazyme Biotech Co., Ltd., Nanjing, China) according to the manufacturer’s instruction. Then two cloned genes were linked to pYES2 expression vector with specific primers (Table 1). After validation of sequences, the empty pYES2 plasmid, recombinant pYES2−*MsUGT003* and pYES2−*MsUGT024* were transformed into a specific yeast *Saccharomyces cerevisiae* strain INVSc1. Then the yeasts were cultivated in a liquid medium containing 2% synthetic complete (SC)−Ura galactose. Further, the yeasts were collected for drought and ABA treatments. The cells were resuspended in 30% PEG-6000 or 250 μM ABA, then the prepared yeast culture was diluted 10 fold and grown on a solid medium containing glucose for 2–3 days to check the expression of the binding protein.

## 4. Discussion

Plant UGTs is the largest glycosyltransferase family that regulates glucose metabolism, homeostasis and participates in detoxification [43]. They play an essential role in plant growth, development and coping with environmental changes by regulating glucose metabolism, homeostasis, and secondary metabolites [32,43]. The *UGT* multigene family has been profiled in many plant species including *A. thaliana* [13], *T. aestivum* [16], *Z. mays* [14], *Linum usitatissimum* [44], and legumes [18], including *M. truncatula*, *M. albus*, *T. paratense*, *Lotus japonicas*, *Glycine max* and *Phaseolus vulgaris.* However, the *UGT* family in alfalfa has not been comprehensively analyzed so far. In this study, a systematic analysis was conducted in the alfalfa *UGT* gene family, including phylogenetic relationships, gene location, conserved motifs, intron/exon position, gene duplication and gene expression.

A recently published genome sequence of alfalfa has provided an opportunity to investigate the diversity in the alfalfa *UGT* multigene family in a great detail. In the present study, we identified 409 *MsUGT* genes. It is worth noting that the number of the alfalfa *UGT* genes was larger than in any plant studied so far, such as *A. thaliana* (120) [13], *Gossypium hirsutum* L. (274) [45], *G. max* (242) [46], *M. albus* (189) [19] and *M. truncatula* (243) [18]. This is probably due to the genome assembly of 32 chromosomes of alfalfa and its large genome size (3068 Mb). A phylogenetic tree was usually constructed for comparing the gene family members and identifying their similarities and differences [47]. The phylogenetic tree displayed that 409 MsUGTs have clustered into 13 groups. The number of UGT groups in different plant species varied largely, *A. thaliana*, *G. hirsutum*, *M. truncatula*, *Cajanus cajan* were clustered into 14, 10, 11, and 15 groups, respectively [13,18,45,47]. Alfalfa UGTs lacked conserved A, G, J, and N groups, however, the number of groups O and P, which were newly identified in *T. aestivum* [16] had 10 and 14 members, respectively. Another newly identified group R in *C. sinensis* [48] had 2 members in alfalfa. Previous studies indicated that group E contained the largest UGT members in most plant species. Our research showed that group E was the third largest group, containing 62 genes. However, group I has expanded to become the largest group, containing 134 genes and composing 32.8% of putative UGT genes in alfalfa. There are *MsUGT* genes clustering in group I in chromosome 6 and chromosome 7. By contrast, there are no gene clusters in group E in alfalfa. The expansion of group I in alfalfa would contribute to its evolution in adapting to different stress conditions. As previously reported, UGTs belonging to group I, such as *UGT83A1* in *O. sativa*, was demonstrated to be induced by abiotic stresses and catalyze the glycosylation of flavonoid to improve plant tolerance [49].

The analysis of chromosome location indicated that *MsUGT* genes were unevenly distributed on 32 chromosomes and mainly clustered on chr6_1, chr6_2, chr6_3, chr6_4, chr7_1 and chr7_2. The gene duplication contributed to the evolutionary novelty and genome complexity by favoring an expanded accumulation of new molecular activities [50,51]. There were 746 pairs of segmental duplications in the *UGT* gene family in alfalfa, suggesting that gene duplication played an essential role in the active expansion and evolution of the *UGT* family in alfalfa. The intron mapping of 409 *MsUGT* genes showed that 65.8% of members contained introns, which is more than the number (42%) of *A. thaliana UGT* genes [13], while close to the number (60%) of *L. usitatissimum* [44], indicating that alfalfa processed UGT gene evolutionary diversity. All MsUGT sequences process motif 1, which contains the UGT conserved PSPG box. Besides, group E contains 16 motifs and the largest group I contains 18 motifs, indicating the expansion of motif numbers would function to improve the stress tolerance of alfalfa. In addition, most gene members belonging to the same subfamily possess comparable motifs and share similar exon-intron patterns in terms of intron members or lengths. These results may provide useful information of the evolution and function of MsUGTs.

Plant UGTs, as enzymes for glycosylation, function in many processes that participate in plant growth and abiotic stress. To get a further understanding of the function of UGTs in alfalfa, the expression pattern in different tissues and abiotic stresses were analyzed, based on the online universal microarray data. The results revealed that 407 genes (99%) were expressed at least in one tissue. Similar patterns were found in *Z. mays* and *L. usitatissimum*, wherein 82% and 73% of *UGT* genes showed an expression pattern at least in on tissue [14,44], In addition, 384 *MsUGT* genes (94%) were expressed at least in one stress. UGTs participate in the glycosylation of substrates, impacting their water solubility, biological activity, subcellular localization and transport characteristics, thereby maintaining metabolic balance in plant cells to improve drought resistance [52]. In alfalfa, a genome-wide association study showed that the *UGT* gene was associated with forage quality under conditions of water deficit [53]. In this study, the role of twelve genes expressed highly in drought stress and ABA signaling were verified by qRT-PCR. The expression of *MsUGT* genes in shoots and root varied differently exposed to PEG and ABA treatments, indicating these representative *MsUGT* genes might have universal functions to drought stress and ABA signaling.

Moreover, we cloned *MsUGT003* and *MsUGT024* to transform into yeasts and confirmed their function under ABA and drought treatments, respectively. The results showed that *MsUGT003*-transformed yeasts appeared more tolerant to drought stress, which was consistent with the results of qRT-PCR analysis. As previously reported, *UGT76E11*, which belongs to group E in *A. thaliana*, modulated the flavonoid metabolism and enhanced the scavenging capacity for ROS to improve the drought resistance [54]; here, *MsUGT003*-transformed yeast gain more tolerance to drought condition, probably due to the positive regulation of *MsUGT003* in drought stress. In addition, *MsUGT024*-transformed yeasts appeared to have more tolerance to ABA treatments. ABA accumulates when plants are exposed to drought and plays an important role in the reduction of water loss by transpiration under water stress conditions. UGTs, glycosylated ABA to ABA-GE, which is a storage form end product of ABA [55]. When the drought stress ceased, the concentration of ABA returned to the normal level rapidly through the glycosylation of UGT to form ABA-GE [24]. The ABA tolerance of the *MsUGT024*-transformed yeast indicates *MsUGT024* probably plays an important role in ABA metabolism in alfalfa. By contrast, the *MsUGT003*-transformed yeast appeared less tolerant to ABA treatment and *MsUGT024*-transfomed yeast showed less tolerance to drought stress. Therefore, the interception of UGT and ABA as well as the specific functions of UGTs in alfalfa under drought stress need further investigation.

## 5. Conclusions

In the present study, the *UGT* gene family of alfalfa was analyzed systematically and comprehensively. *UGT* genes (409 in number) were identified and their phylogenetic tree, chromosomal location, duplication events, and exon-intron structures, conserved motifs and cis-regulatory elements were evaluated to get a better insight into the role of the *UGT* gene family in alfalfa. The RNA-seq analysis confirmed that *MsUGT* genes expressed in different tissues and under different abiotic stresses, qRT-PCR further confirmed the results of digital expression analysis observed under drought and ABA treatments. Heterologous expression in yeasts indicated that *MsUGT003* and *MsUGT024* were in response to drought stress and ABA signaling. To sum up, our study could have high importance in exploiting the potential molecular function of *MsUGT* genes in drought stress and in applying molecular approaches in the breeding of alfalfa, but their functions still require a series of experiments to confirm this.

## Figures and Tables

**Figure 1 ijms-23-07243-f001:**
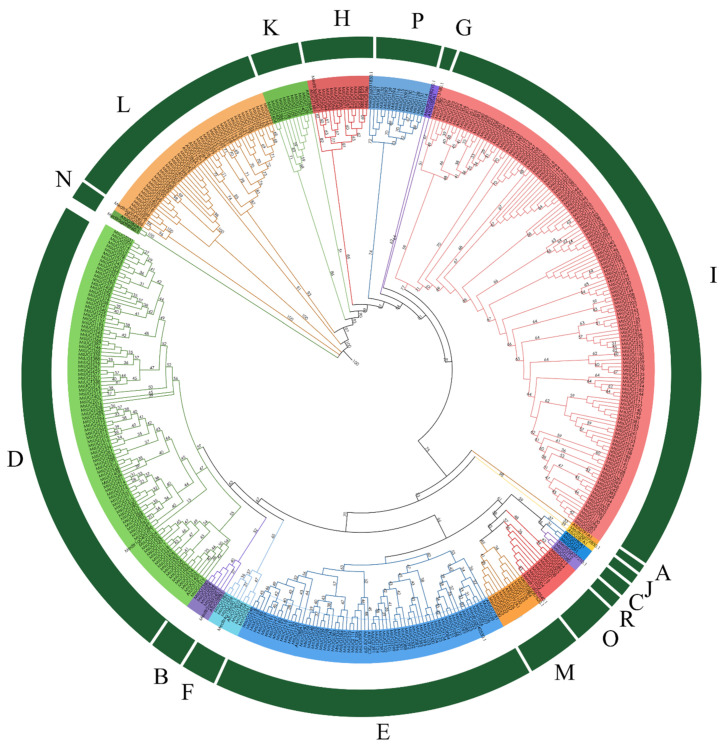
Phylogenetic tree of UGTs from alfalfa, *A. thaliana* and *M. truncatula*. 17 *A. thaliana* UGTs members and 14 *M. truncatula* and *409* identified UGTs from alfalfa were contracted the phylogenetic trees using the MEGA 7 program. These members were clustered into 17 groups (A–N, and O, P, R), and indicated by different colors.

**Figure 2 ijms-23-07243-f002:**
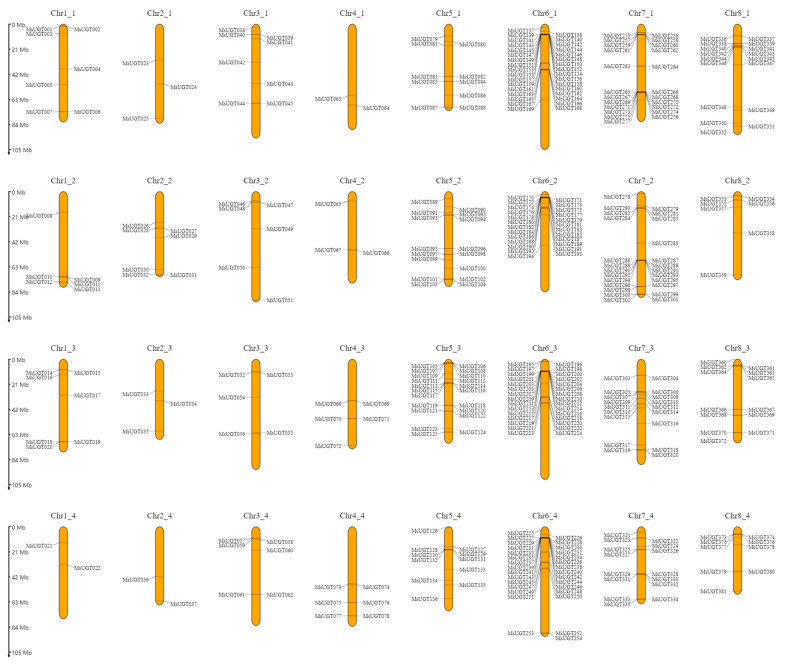
The physical location of *MsUGT* genes across the 32 chromosomes in the alfalfa genome. These genes were named *MsUGT001* to *MsUGT409* according to the physical distribution on the chromosomes. The orange bars represent 32 chromosomes, and the position of each *MsUGT* gene is indicated by the black lines.

**Figure 3 ijms-23-07243-f003:**
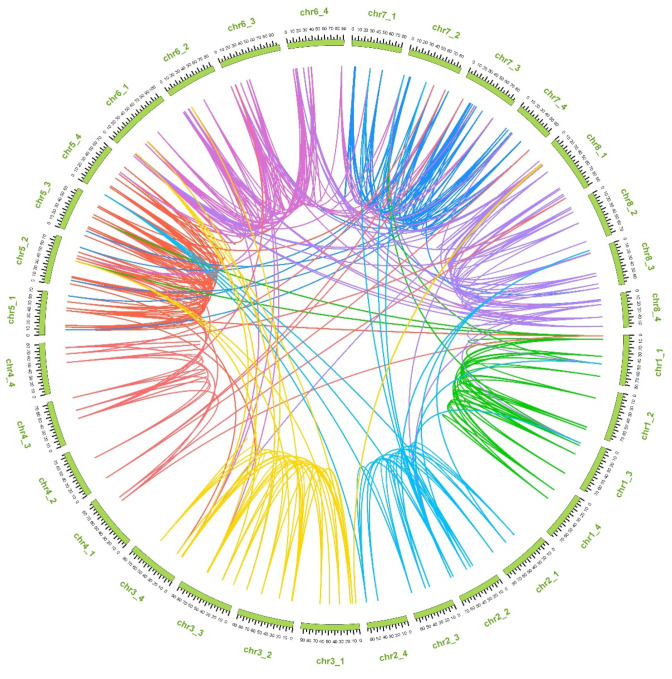
Synteny analysis of the *UGT* genes of alfalfa. Chromosomes are shown in the outer circle in green. Eight different color lines inside indicate duplicated *MsUGT* gene pairs in 8 chromosomes.

**Figure 4 ijms-23-07243-f004:**
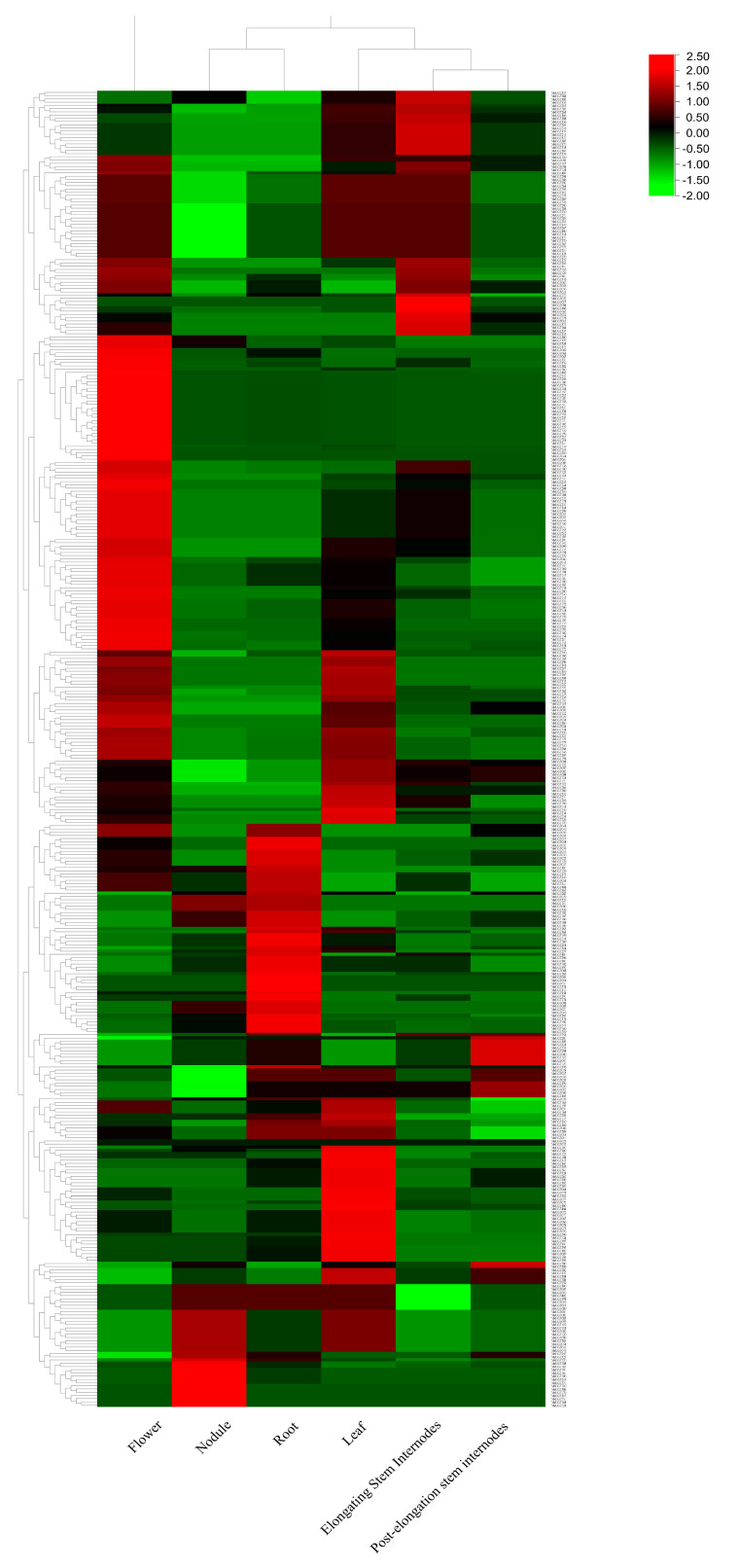
Expression profile of *MsUGT* genes in six different tissues (flower, nodule, root, leaf, elongating stem internodes, post−elongation stem internodes). The data were retrieved from transcriptome datasets. The heatmap was constructed by TBtools software (v1.098745, Chen, C. J., Guangzhou, China) and expression values (FPKM) were normalized.

**Figure 5 ijms-23-07243-f005:**
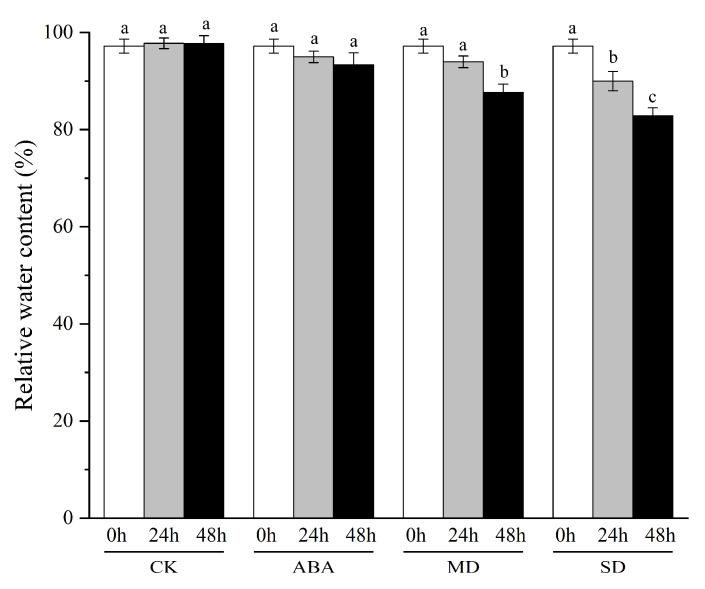
The relative water content of leaves under PEG induced drought stress and ABA treatments. CK, MD, SD and ABA represent the control condition, mild and severe drought and ABA treatments, respectively. The error bars indicated the standard errors of three biological replicates. Lowercase letters indicate significant differences.

**Figure 6 ijms-23-07243-f006:**
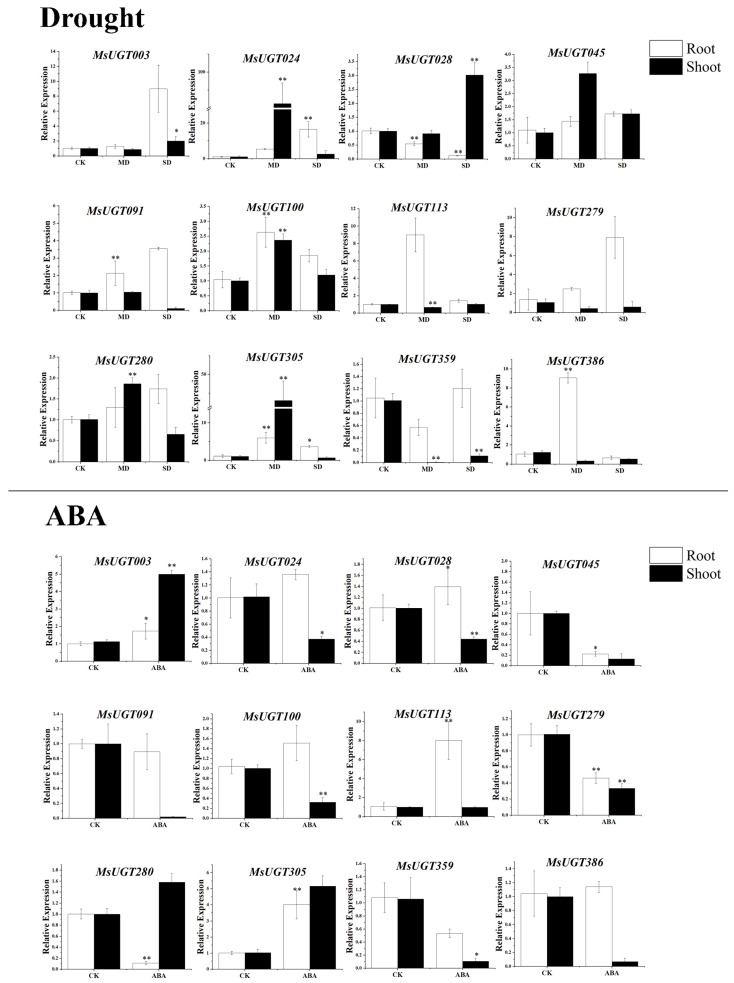
qRT-PCR results of the relative expression of twelve selected *MsUGT* genes in response to drought stress and ABA treatment in shoot and root. CK, MD, SD and ABA represent the control condition, mild and severe drought and ABA treatments, respectively. Asterisks indicate the significance compared with CK; * represents significant (*p* < 0.05) and ** represent highly significant (*p* < 0.01). The error bars indicate the standard errors of three biological replicates.

**Figure 7 ijms-23-07243-f007:**
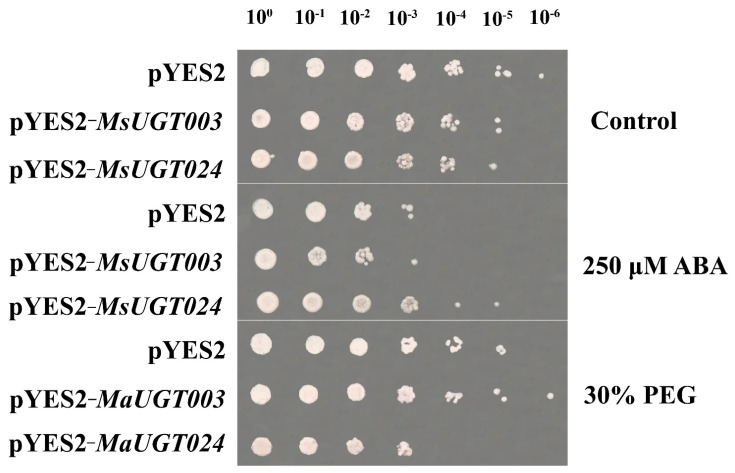
Drought stress tolerance and ABA treatment analysis of *MsUGT003* and *MsUGT024* genes in heterologous expression in yeast with the comparation of empty pYES2 (control) line. The serial dilutions (10^0^, 10^1^, 10^2^, 10^3^ 10^4^, 10^5^, 10^6^) of yeast cultured under control, 30% PEG or 50 μM ABA conditions were spotted onto solid medium containing SC−Ura.

**Table 1 ijms-23-07243-t001:** Primers used in this study.

Primer Name	Sequences
MsUGT280-F	TTTGTCCACAAGAACCACTATGAG
MsUGT280-R	TCCTTGGAAGACCACTTCGAG
MsUGT305-F	TTGATTCTGAAGGAAATCCCAC
MsUGT305-R	AATTGGTAGGAACTTTGGTGAC
MsUGT386-F	ACTGACCAACCAACAAATGCT
MsUGT386-R	TGCTAGAACTACCATCCTCCT
MsUGT045-F	TCTTACCCTCACTTCACTTTCC
MsUGT045-R	CCTCCATTCGAGATTCAGTCCT
MsUGT113-F	CCATTAGAGGAGACGAGGAC
MsUGT113-R	TATTGAGGTGGTGATGAAGAGG
MsUGT279-F	TAATACCGCTCAAAGGATTGCAG
MsUGT279-R	ATGAAGGTAAAGGTCCAATGGT
MsUGT359-F	CTCCAAGGAAGTCCATGATGTC
MsUGT359-R	GTGGTGTTGACAAGTTGAATGG
MsUGT003-F	AAGCTTCTTTACATCTTCGCGAG
MsUGT003-R	CATTTGTGCCATCATGAAATCGAC
MsUGT091-F	CACTTAACATCCACAAACTTTCAC
MsUGT091-R	CCTTTAGTAAGCTCGTCATGG
MsUGT024-F	TTCGGATGAGTTGGGAAGAG
MsUGT024-R	CAGAACTTCCACCTTCTCTAACAG
MsUGT100-F	GGAGCATTTCTAAGTCATTGTGG
MsUGT100-R	CACCCTTCTCTATTGTTGCCTC
MsUGT028-F	TTTCTTCCATTTCCATCTCCC
MsUGT028-R	GTATCCTAAATTGTTGTCACTGTC
MsMsUGT003-pYES2-F	cttggtaccgagctcggatccATGGTTCCGTCTTTTGAAG
MsMsUGT003-pYES2-R	tacatgatgcggccctctagaCTATCTAGTAATATGAGCGATGAAAG
MsMsUGT024-pYES2-F	cttggtaccgagctcggatccATGACGTTTCAACCAGGC
MsMsUGT024-pYES2-R	tacatgatgcggccctctagaCTAAAATTGTAACACAAGTTGTTC

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
