# Peer review of "Genome-Wide Analysis and Profile of UDP-Glycosyltransferases Family in Alfalfa (Medicago sativa L.) under Drought Stress"

_ijms, 2022, doi:10.3390/ijms23137243_

Round 1

Reviewer 1 Report

Dear Authors,

I think that manuscript titled “Genome-wide analysis and profile of UDP-glycosyltransferases family in alfalfa (Medicago sativa L.) under drought stress” address an important issue of better understanding the drought stress in Medicago sativa. Both the number and the quality of conducted analysis are well done and are a great asset of this manuscript.

However, the main problem is selection of stress conditions. The justification for the selection of stress condition was written superficially. There is only short information in Material and Methods chapter about this condition. Author did not put any attention to justify the selection in the introduction.

Moreover Authors did not show any data what effects the stress factors had on the seedlings. How it change the water potential/water content/ seedlings shape and size? They limited the description of the strength of stress to the statement of moderate and strong stress which, without the provided data, say nothing at all.

Secondly the methodology of application and stress caused by ABA is also unclear. It is also unclear why authors decided to use concentration 100 μM ABA solution? Why not bigger or smaller? Did authors have any information what was the final solutions of ABA in tissues of different part of seedlings i.e. leaves, shoot, root after spray with solution of 100 µM ABA?

What mean sprayed onto the whole plant. Should I understand that the plant was taken out from growing medium then whole plant (leaves, shoot and root) was sprayed and then the seedling lay for 48 hours before sampling? Whether only the above-ground part of the seedlings was sprayed? Why authors did not add ABA to the growing medium as in regular stress conditions drought stress signals transmitted through ABA are going from root to shoot and then to leaves?

In my opinion also discussion should be improved as it is mainly descriptive. There is lacking of conclusion. I will show only few examples:

Line 349-355

‘Previous studies indicated that group E contained the largest UGT members in most plant species. Our research showed that group E was the third largest group, containing 62 genes and composing 15.2% of putative UGT genes in alfalfa. Plenty of UGT genes classified into group E have been functionally characterized, including glycosylation of small molecule volatile com- pounds, and synthesis of secondary metabolites[45, 46]. However, group I has expanded to become the largest group in alfalfa, containing 134 genes.”

Authors did not discussed why group I has expanded to become the largest group in alfalfa, contrary to group E which is largest in most investigated species?

What is the meaning of this information? How this influence the drought tolerance of alfalfa in comparison to different species i.e. this with the biggest E group? It is relevant or not? How many motifs contain Group I? Since we know that group E contained 16 motifs? If the numbers of motifs have influence on ABA or drought tolerance?

Interestingly at the end of the introduction Authors put very important information.

“MsUGT003-transformed yeasts appeared more tolerant to PEG and MsUGT024-transformed yeasts was more tolerant to ABA treatment,’”

However also this information in my opinion is insufficient discussed. What is important to notify according to Figure 6 MsUGT003-transformed yeasts are less tolerant o ABA treatment and MsUGT024-transformed yeasts are less tolerant to PEG. How authors could explain this situation with suggested role of MsUGT genes in drought/ABA stress response?

Taking into account all the above considerations, I believe that the manuscript under the title “Genome-wide analysis and profile of UDP-glycosyltransferases family in alfalfa (Medicago sativa L.) under drought stress” can be accepted for publication by International Journal of Molecular Science after resubmission.

Author Response

Thanks a lot for your kind suggestions. The authors are grateful for the valuable and helpful comments of the reviewers. 

Reviewer 2 Report

This paper is focused on UDP glycosyltransferases (UGT) in alfalfa.  The authors analyzed the alfalfa genome for the total number of UGT's and looked into some of the UGTs for expression under drought conditions. 

Line 74: I suggest you add the work "nicknamed" before the Queen of Forages line.

Figures 1-3 are good

Figure 4 needs labels on which tissue type go with the different columns.

Figure 5 : You need to define MD, SD and CK, this is not defined in the entire article (that I could find).

Line 339 there are two spaces between any and plant

Line 341: Is allele aware used in the proper context here?

Line 349:  The sentence 'Another newly identified group R in C. sinensis [44] had 2 members in alfalfa.' is in a smaller font size compared to the rest of the paper. Please fix

Line 409 Add "respectively"... to help define MsUGT003 and 024 to the different treatments.

Author Response

(The authors gave the same response as above.)

Round 2

Reviewer 1 Report

The revised version is fine for me so it could be published in IJMS.